

# Metabolomics and proteomics analyses reveal the role of the glycerophospholipid metabolism pathway in unexplained recurrent spontaneous abortion

Yihong Chen[1,2,*], Xiumei Zhao[2,*], Bei Gan[2], Leiyi Jin[2], Juanbing Wei[2] and Jianying Yan[1]

[1] College of Clinical Medicine for Obstetrics & Gynecology and Pediatrics, Fujian Medical University, Department of Obstetrics and Gynecology, Fujian Maternity and Child Health Hospital, Fujian Clinical Research Center for Maternal-Fetal Medicine, Fuzhou, China
[2] Department of Obstetrics and Gynecology, The First Affiliated Hospital, Fujian Medical University, Fuzhou, China
[*] These authors contributed equally to this work.

Corresponding authors
Juanbing Wei, weijuanbing@126.com
Jianying Yan, lwjb1968@163.com

## ABSTRACT

**Background**. Unexplained recurrent spontaneous abortion (URSA) is a complex pregnancy complication with a high miscarriage rate. Incomprehensive understanding of the molecular mechanism in URSA also leads to a lack of effective treatment methods. Hence, the current study aimed to explore the underlying pathogenesis of URSA applying metabonomic and bioinformatics analysis.

**Methods**. The decidual tissues of eight URSA samples and eight normal pregnancy (normal control, NC) samples were collected for liquid chromatography-mass spectrometry (LC-MS) analysis using the Progenesis QI metabolomics software. The orthogonal partial least squares discrimination analysis (OPLS-DA) and the Human Metabolome Database (HMDB) were employed for differential metabolite analysis and pathway enrichment analysis, respectively. Kyoto Encyclopedia of Genes and Genomes (KEGG) pathway topological analysis was performed to rank the importance of pathways involved in URSA, and differential proteins were identified based on fold change difference. Finally, a metabolic network was visualized by the Cytoscape tool.

**Results**. After LC-MS analysis and quality control, samples in the same group showed high consistency and reliability. Differential metabolites between NC and URSA groups were mainly enriched to five biological processes, with glycerophospholipid metabolism pathway containing the greatest number of differential metabolites. KEGG enrichment analysis showed significant differences in glycerophospholipid metabolism, bile secretion, and choline metabolism pathways, with glycerophospholipid metabolism showing a higher pathway importance. Proteome and metabolome analysis revealed a total of 65 overlapping pathways involved in the differential proteins and differential metabolites, and finally *PLD1*, *CHPT1* and *PLA2G2A* were identified as the key genes in glycerophospholipid metabolism pathway.

**Conclusion**. LC-MS analysis revealed that glycerophospholipid metabolism pathway and its three key genes were crucially involved in URSA progression, providing novel insights into the treatment strategy of URSA.

## INTRODUCTION

Recurrent spontaneous abortion (RSA) has a complex etiology and lacks specific clinical manifestations. RSA is defined as two or more clinically consecutive or discontinuous miscarriages of pregnancy occurring before 20 weeks of gestation, affecting 1–5% of women of reproductive age (*Practice Committee of the American Society for Reproductive Medicine, 2013*). Several etiologies of RSA have been proposed, such as pathogens infection, endocrine dysfunction, anatomical abnormalities, thrombosis, chromosomal abnormalities, immunological factors and environmental factors (*Alijotas-Reig & Garrido-Gimenez, 2013*). However, more than half of RSA cases with unclear pathogenesis is known as unexplained RSA (URSA) (*Wilczyński et al., 2012*). Currently, one study reported that URSA is closely associated with immunological imbalance (*Alijotas-Reig, Melnychuk & Gris, 2015*). From an immunological point of view, a healthy pregnancy can be regarded as a successful allogeneic transplant, during which the embryo expresses the semi-allogeneic antigens of the father, which suggests that the establishment of an immune-tolerant embryo in the maternal immune system is essential for a successful pregnancy.

The decidua and its residing immune cells contain a specific immune cell complement that comprises 70% of natural killer (NK) cells, playing a crucial role in maintaining pregnancy through invading the uterine mucosa and controlling the invasion of trophoblasts during the first trimester of pregnancy (*King et al., 1991*). Endometrium with insufficiently developed decidua or a lack of decidua will cause the failure of embryo implantation, manifesting as the inhibition of cell proliferation, apoptosis of decidua cells, and abnormal expression of decidualized genes (*Zhao et al., 2022*). Therefore, the immune microenvironment of decidua has a crucial significance for embryo implantation and development (*Hong et al., 2021*). For example, the proportion of Th2 cells to maintain allograft immune tolerance for a normal pregnancy increases, while the number of Th1 cells that secrete Th1-type cytokines (tumor necrosis factor-α (TNF-α), interferon-γ and interleukin-2) decreases (*Wu et al., 2015*; *Rowaiye et al., 2021*). Notably, the imbalance of Th1/Th2 ratio can lead the susceptibility to pathological pregnancy (*Chaouat, 2007*). The decidual NK cells phenotypically and functionally differ from blood NK cells, contributing to vascular remodeling and trophoblast invasion through secreting a series of regulatory molecules, cytokines and chemokines (*Kalkunte et al., 2009*). Study also showed that significantly downregulated β-catenin in URSA patients affects trophoblast proliferation (*Li et al., 2015*). In recent years, the pathogenesis of URSA has been extensively explored from various aspects, such as vaginal dysbiosis and nicotinamide adenine dinucleotide (NAD) deficiency (*Grewal et al., 2022*; *Cuny et al., 2020*). It has been found that miRNA-125b suppresses the expression of TNF-α (*Tili et al., 2007*). *Li et al. (2020)* screened 123 differentially expressed circRNAs that affect URSA progression. However, these findings

for understanding the etiology of URSA are still limited. The function of proteins is regulated by transcription, translation and post-modification, and the changes of certain proteins in metabolism are decisive in the occurrence of some diseases. Compared to transcriptomics, quantitative metabonomic technique is also a useful tool to explore the differential metabolites associated with key disease processes (*Hong et al., 2021*), facilitating the discovery of potential biomarkers and improving the understanding of molecular mechanisms underlying URSA.

This study collected the decidual tissues from URSA and normal pregnancy samples, and performed liquid chromatography-mass spectrum (LC-MS) analysis. Principal component analysis (PCA) and orthogonal partial least squares discrimination analysis (OPLS-DA) with variable importance in projection (VIP) value were utilized to screen differential metabolites showing intergroup significance. Next, pathway enrichment analysis of metabolites was conducted based on the Human Metabolome Database (HMDB) and the Kyoto Encyclopedia of Genes and Genomes (KEGG) database, and the pathway importance was ranked by KEGG pathway topological analysis according to the position of metabolites in the pathway network. In addition, analysis on differential proteins between the URSA and NC groups identified glycerophospholipid metabolism as a key pathway in URSA progression based on the intersection of differential metabolites and proteins. The present study discovered specific marker molecules for URSA, improving the understanding of the pathogenesis of URSA and laying a solid foundation for the prevention and treatment of the disease.

## MATERIALS AND METHODS

### Metabolomics analysis and data acquisition

The decidual tissues of URSA women ($n = 8$) who had two or more consecutive spontaneous abortions with the same spouse before 10 weeks of gestation were collected, while excluding those who had genetic or endocrine termination of pregnancy. The decidual tissues derived from women with normal pregnancy but without history of spontaneous abortion, abnormal birth or other diseases (gestational age 6–10 weeks) served as normal control (NC) samples ($n = 8$). The exclusion criteria included irregular menstrual cycles, genital infections, abnormalities of uterine cavity, antiphospholipid syndrome, thrombotic disorders, chronic hypertension, diabetes, kidney disorders, thyroid conditions, autoimmune disorders, and cardiovascular diseases, along with chromosomal abnormalities in either partner. All the samples were assessed by histopathologists to determine the secretory phase without the interference from hormonal medication. The histopathologists were blinded to the clinical information of the patients during the evaluation in order to avoid bias. All the experimental procedures were approved by Ethics Committee of First Affiliated Hospital of Fujian Medical University (Approval No. MTCA, ECFAH of FMU [2015] 084-2), and the patients enrolled signed the informed consent.

For metabolomics analysis, approximately 50 mg of tissue was homogenized and lysed using sonication at 30 kHz, followed by cryogenic ultrasound extraction for 30 min (min, at 5 °C, 40 kHz) using 400 μL of extraction solution (methanol: water = 4:1, v/v) containing

0.02 mg/mL of internal standard L-2-chlorophenylalanine. After standing for 30 min at −20 °C, the centrifugal supernatant (15 min, 13,000 g) was applied for LC-MS analysis using an ultra-high-performance LC (UHPLC)-Q Exactive HF-X system (Ultimate 3000LC, Thermo, Waltham, MA, USA). Each experimental sample was filtered through a 0.22 μm membrane filter, and a final injection volume of 2 μL was loaded into the LC-MS system for analysis. Quality control (QC) of samples was conducted by pooling equal volumes (20 μL) of the supernatant of each experimental sample, and 2 μL of the QC sample was injected for system stability assessment. ACQUITY UPLC HSS T3 column (100 mm × 2.1 mm i.d., 1.8 μm; WatersCorporation, Milford, MA, USA), which consisted of two mobile phases of A (95% water +5% acetonitrile containing 0.1% formic acid) and B (47.5% acetonitrile +47.5% isopropanol +5% water containing 0.1% formic acid), was used to perform chromatography at 0.4 mL/min of column flow rate at 40 °C of column temperature. Different gradient elution programs were applied under positive and negative ion modes to optimize metabolite separation (Table S1). The column was equilibrated at the initial mobile phase composition for 2 min prior to each injection. The total run time for each sample was 8 min, ensuring sufficient separation and reproducibility of analytes under both ionization modes. Furthermore, the detailed mass spectrometry parameters were as follows: scan range 70–1,050 m/z, sheath gas flow rate 50 arb, auxiliary gas flow rate 13 arb, heater temperature 425 °C, capillary temperature 325 °C, S-Lens RF level 50, and resolution 60,000 (Full MS) and 7,500 ($MS^2$). Subsequently, the data were extracted, aligned and identified using metabolomics software Progenesis QI (WatersCorporation, USA) (*Yang et al., 2022*; *Zhang et al., 2023*; *Meng et al., 2024*).

## Sample quality control evaluation

We performed the quality control evaluation and filtered raw matrix before data analysis. The filtering criteria were as follows: (1) Variables with 80% nonzero value were retained; (2) missing value was replaced by the 1/2 minimum value in matrix; (3) the total peak area was normalized after removing the variables with relative standard deviation (RSD) ≥30% in QC samples; (4) Log10 conversion was performed for data analysis. For quality evaluation, the correlation of metabolic data of eight URSA, eight NC and three QC samples was analyzed using the Spearman method to ensure the consistency during the assessment of sample metabolites. PCA was used for the assessment of outliers. The metabolic differences between the control and sample groups were compared by OPLS-DA, and the permutation testing was performed for evaluating the reliability of OPLS-DA employing the "ropls" R package (*How et al., 2023*).

## Screening of differential metabolites

In PLS-DA analysis, the VIP value is a crucial indicator of independent variable in explaining the dependent variable, and can be used to screen variables with the greatest contribution among various groups (*How et al., 2023*). The T test combining the OPLS-DA method was employed to identify metabolites with significant intergroup differences (VIP > 1 and $p$ value < 0.05). The HMDB database (https://www.hmdb.ca/), KEGG database (https://www.kegg.jp/) and LIPID MAPS database (https://www.lipidmaps.org) provided

the KEGG ID and the annotation of differential metabolites. Then, KEGG pathway enrichment analysis of the metabolite set was performed using Fisher's test. Based on the KEGG database, important biomolecules in the pathways were assessed by KEGG pathway topological (MetPA) analysis and betweenness centrality (*Xia & Wishart, 2010*), which was calculated according to the position of each biomolecule in a pathway.

## Differential protein analysis between URSA and NC groups

The decidual tissue samples were lysed with radio immunoprecipitation (RIPA) lysis buffer, and the supernatant was collected by ultrasonic fragmentation and centrifugation at 12,000 g for 15 min at 4 °C. After quantifying the protein concentration with a bicinchoninic acid (BCA) kit (Applygen Technologies Inc., Beijing, China), an aliquot of protein (100 µg) was trypsinized, and the resulting peptides were desalted and enriched using solid-phase extraction (SPE). The peptides were analyzed by an UHPLC system coupled with Q Exactive mass spectrometer in data independent acquisition (DIA) mode and further quantitatively analyzed by the Spectronaut software matching the DIA spectral libraries (*Su et al., 2022*; *Makhmut et al., 2024*; *Krieger et al., 2019*). Next, significantly differentially expressed proteins between the URSA and NC groups (setting fold change (FC) < 0.67 or FC > 1.50) were screened and subjected to gene ontology (GO) enrichment analysis using the hypergeometric distribution method with Fisher test ($p < 0.05$) (*Song et al., 2023*).

## Overlapping pathway analysis and KEGG pathway enrichment analysis

"VennDiagram" R package was used to visualize the intersection between differential metabolites and enriched protein pathways, and the top 10 overlapping pathways with the greatest number of differential metabolites and proteins were selected. KEGG annotation analysis with Fisher test was performed to identify most relevant biological processes (BPs) in protein/metabolite sets to improve the reliability of the research. In addition, we applied the MetScape of Cytoscape tool to visualize the metabolic network of differential proteins and metabolites in the glycerophospholipid metabolism pathway based on the Edinburgh Human Metabolic Network (EHMN) and KEGG COMPOUND Database. Metabolite differences between diseased and normal samples were compared by single sample gene set enrichment analysis (ssGSEA) using the gene set variation analysis (GSVA) R package (*Hänzelmann, Castelo & Guinney, 2013*).

## Cell culture and treatment

The telomerase-immortalized human endometrial stromal cells (T-hESCs, ATCC CRL-4003) were obtained from ATCC (Manassas, VA, USA). Human stromal cells were maintained in Dulbecco's modified Eagle's medium (DMEM)/F12 (Sigma-Aldrich, Darmstadt, Germany) supplemented with 10% fetal bovine serum (FBS) (Biological Industries, Shanghai, China). To promote *in vitro* decidualization, T-hESCs were processed with 500 µM of db-cAMP (dibutyryl cAMP sodium salt, Sigma-Aldrich) and 1 µM of medroxyprogesterone acetate (MPA, Absin, Shanghai, China), following previously established protocols (*Qi et al., 2015*; *Yang et al., 2021*). Si-negative control (si-NC) and si-*CHPT1* were obtained from GenePharma Co (Shanghai, China). Transfection of these

**Table 1  Sequences of primers for qRT-PCR.**

| Gene | Primers (5′–3′) | |
| --- | --- | --- |
| | Forward primers | Reverse primers |
| PLD1 | CCCAGCGATCCCAAGATACAA | GACAGCCGGAGAGATACGTCT |
| CHPT1 | CACCGAAGAGGCACCATACTG | CCCTAAAGGGGAACAAGAGTTTG |
| PLA2G2A | ATGAAGACCCTCCTACTGTTGG | GCTTCCTTTCCTGTCGTCAACT |
| PRL | GGAGCAAGCCCAACAGATGAA | GGCTCATTCCAGGATCGCAAT |
| IGFBP1 | TTGGGACGCCATCAGTACCTA | TTGGCTAAACTCTCTACGACTCT |
| GAPDH | TGAGTATGTCGTGGAGTCTA | CACAAAGTTGTCATTGAGAG |

plasmids into cells was conducted with the use of Lipofectamine 3000 reagents (Invitrogen, Carlsbad, CA, USA). The transfection efficiency was assessed after 48 h using qRT-PCR.

## Quantitative real-time PCR (qRT-PCR)

Total RNA was isolated using TRIzol (Invitrogen, USA), and cDNA was generated with a SuperScript™ III reverse transcriptase kit (Invitrogen, USA). Quantitative real-time polymerase chain reaction (QRT-PCR) was carried out utilizing SYBR Green qPCR Master Mix (Takara, Shanghai, China) on an ABI Prism 7500. The qPCR protocol began with an initial denaturation at 94 °C for 30 seconds (s), followed by 40 amplification cycles at 94 °C for 5 s and at 60 °C for 30 s. The relative expressions of *PLD1*, *CHPT1*, *PLA2G2A*, *PRL*, and *IGFBP1* were calculated using the $2^{-\Delta\Delta Ct}$ method, with *GAPDH* as an internal control. All the experiments were conducted in triplicate. The sequences of all primers are listed in Table 1.

## Cell counting kit-8 (CCK-8) assay

Cells at the logarithmic phase were plated into a 96-well plate at a concentration of $1 \times 10^4$ cells/well and incubated at 37 °C with 5% $CO_2$ for 0, 24, 48, and 72 h. Subsequently, 10 μL of CCK-8 solution was added to the medium and the samples were maintained at 37 °C for 2 h. To generate cell counting kit-8 (cck-8) curve, the absorbance measured at 450 nm was plotted on the $y$-axis, while time was indicated by the $x$-axis.

## Flow cytometry

The decidualized T-hESCs transfected with *CHPT1*-specific siRNA (si-*CHPT1*) or si-NC were collected, rinsed in phosphate buffered saline (PBS), and then resuspended in 195 μL of Annexin-V fluorescein isothiocyanate (FITC) (BD Biosciences, Franklin Lakes, NJ, USA) containing 5 μL of propidium iodide (PI), according to the instructions. Subsequently, the cells were incubated in the dark at room temperature for 10 min and then subjected to flow cytometry. The data were analyzed using Lysis software (EPICS-XL, Ramsey, MN, USA).

## Statistical analysis

The R software and GraphPad Prism software were applied for data analysis and visualization. Spearman method was used for correlation analysis between two variables, and the Fisher test and unpaired $t$-test were used for calculating significant differences, with a $p < 0.05$ denoting a statistical significance. (*$p < 0.05$, **$p < 0.01$, ***$p < 0.001$).

## RESULTS

### Samples with a high repeatability for differential metabolite analysis

A total of 8 USRA, 8 NC and 3 QC samples were used to perform LC-MS analysis. Correlation analysis of these samples showed that the metabolites in the same groups exhibited a high consistency. URSA (1, 3–8), QC (01–03) and NC (1–8) were respectively clustered into the URSA, QC and NC groups with a high correlation (Fig. 1A), and only samples in the same group with the smallest variation were included in further differential metabolite analysis. Then, PCA was performed to identify abnormal samples, and the results showed that the samples in same groups were clustered together (Fig. 1B), indicating a high repeatability of QC and a stable analytic system. OPLS-DA analysis revealed that the samples were classified into NC and URSA groups with a clear boundary (Fig. 1C). Further permutation testing indicated the intercept value of $Q^2$ was less than 0 (Fig. 1D), excluding contingency and suggesting a strong imitative effect of OPLS-DA model.

### Glycerophospholipid metabolism was a key pathway in URSA progression

After student' $t$ test and OPLS-DA analysis, differential metabolites between NC and URSA groups were obtained. Specifically, the metabolites were implicated in the KEGG pathways, such as organismal systems, metabolism, human diseases, environmental information processing and cellular processes (Fig. 2A, Table S2), with the most differential metabolites involved in the glycerophospholipid metabolism pathway (Fig. 2B). In addition, the KEGG enrichment pathway showed that glycerophospholipid metabolism, bile secretion, and choline metabolism in cancer pathways were the significantly different pathways between NC and URSA groups ($p < 0.05$, Fig. 2C). The pathway importance score was calculated by hypergeometric test, and it was found that the top five pathways affecting the progression of URSA were steroid degradation, furfural degradation, glycerophospholipid metabolism, caffeine metabolism and steroid hormone biosynthesis (Fig. 2D).

### Glycerophospholipid metabolism involving its biosynthesis for URSA progression

A total of 115 upregulated and 227 downregulated proteins with differential expression between NC and URSA groups were screened (Fig. 3A, Table S3). The GO function enrichment analysis of differential proteins in the NC group showed that these proteins were associated with structural molecule activity, response to stimulus, protein-containing complex, molecular function regulator activity, localization, developmental processes, biological regulation and binding process (Fig. 3B), while these differential proteins in the URSA group were related to coronavirus disease-COVID-19, regulation of actin cytoskeleton, neutrophil extracellular trap formation, extracellular matrix (ECM)-receptor interaction and glycerophospholipid biosynthesis-lacto and neolacto series process (Fig. 3C). These results indicated that glycerophospholipid metabolism played a crucial role in URSA progression.
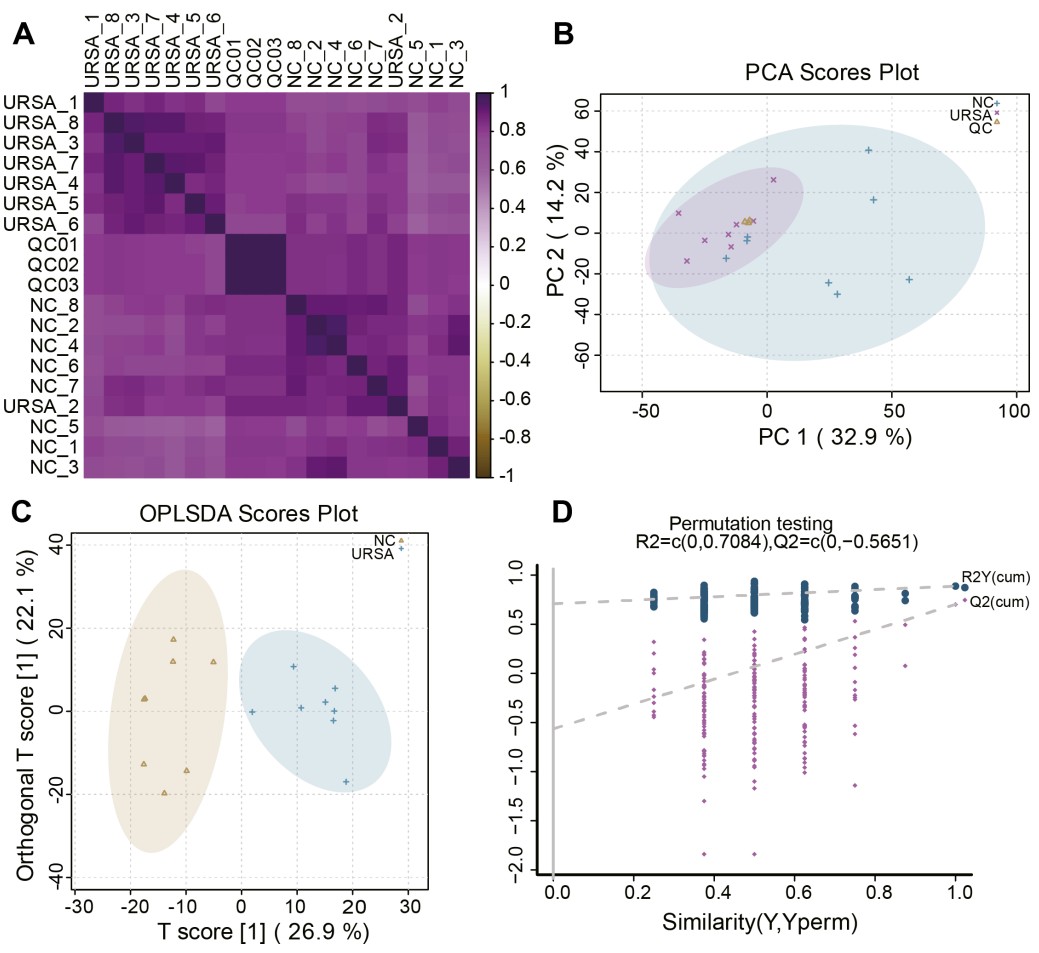

**Figure 1** **Sample reliability detection for metabolomics analysis.** (A) Heatmap of the sample correlation. (B) Principal components analysis (PCA) for outlier. (C) OPLS-DA for differential metabolites analysis. (D) Permutation testing for the reliability of OPLS-DA.

## Pathway enrichment analysis supported the key role of glycerophospholipid metabolism in URSA

Proteome and metabolome analysis revealed 65 overlapping pathways involved in the differential proteins and differential metabolites (Fig. 4A). The top 10 pathways containing the most differential proteins and metabolites are shown in Fig. 4B, in particular, the glycerophospholipid metabolism pathway contained seven differential proteins and 13 metabolites. The KEGG enrichment analysis of the differential proteins and metabolites in each group showed that glycerophospholipid metabolism, pantothenate and coenzmye A (CoA) biosynthesis and drug metabolism-cytochrome P450 were significantly enriched (Fig. 4C). These results further indicated that the glycerophospholipid metabolism was an important pathway in mediating URSA development.

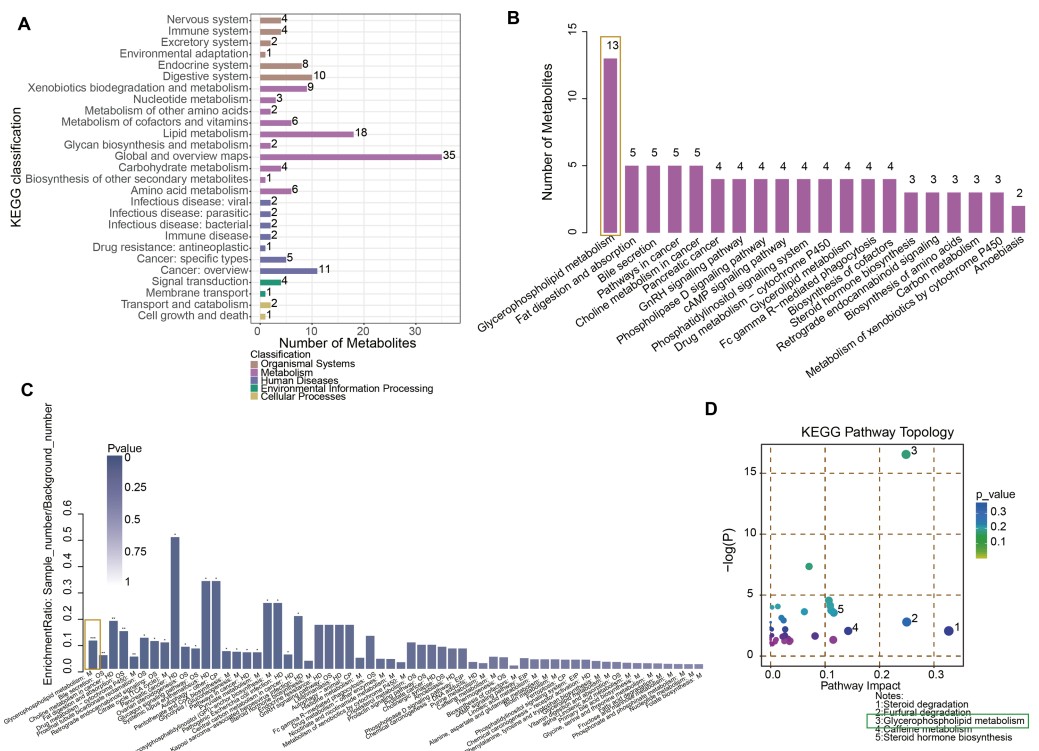

**Figure 2** **Functional enrichment analysis of metabolites.** (A) KEGG enrichment analysis of differential metabolites. (B) The number of metabolites in different pathways. (C) Pathway difference analysis between NC and URSA by using hypergeometric test. (D) Pathway importance score analysis.

## The key metabolites involved in the dysregulation of glycerophospholipid metabolism

Conjoint analysis revealed that the differential proteins and metabolites were enriched in glycerophospholipid metabolism pathway. Thus, these proteins and metabolites were extracted to perform Spearman correlation analysis. The results showed that the proteins of P80404, Q13393 and Q8WUD6 were positively correlated with the production of most metabolites, including PE (phosphatidyl ethanolamine), CL (cardiolipins), PC (phosphatidylcholine) and PS (phosphatidylserine), while the proteins of Q9UKY3, P14555 and P50416 were positively correlated with CL (i-12), phosphatidic acid (PA) and N-methyl phosphatidylethanolamine (PE-NMe) (Fig. 5A). Specifically, CL (i-12) refers to cardiolipin containing an iso-12 branched-chain fatty acid, a key component of mitochondrial membranes involved in energy metabolism and apoptosis regulation (*Banthiya et al., 2016*; *Paradies et al., 2014*). PE-NMe refers to N-methyl phosphatidylethanolamine, an intermediate product in the glycerophospholipid metabolism pathway and a precursor for phosphatidylcholine synthesis, which could potentially influence membrane fluidity and metabolic activity (*Vance, 2008*; *Jones & George, 2013*). Analysis on the metabolic network showed that *PLD1*, *CHPT1* and *PLA2G2A* genes encoded Q13393, Q8WUD6 and P14555 proteins (blue) to regulate the enzyme activity of phospholipase D, diacylglycerol

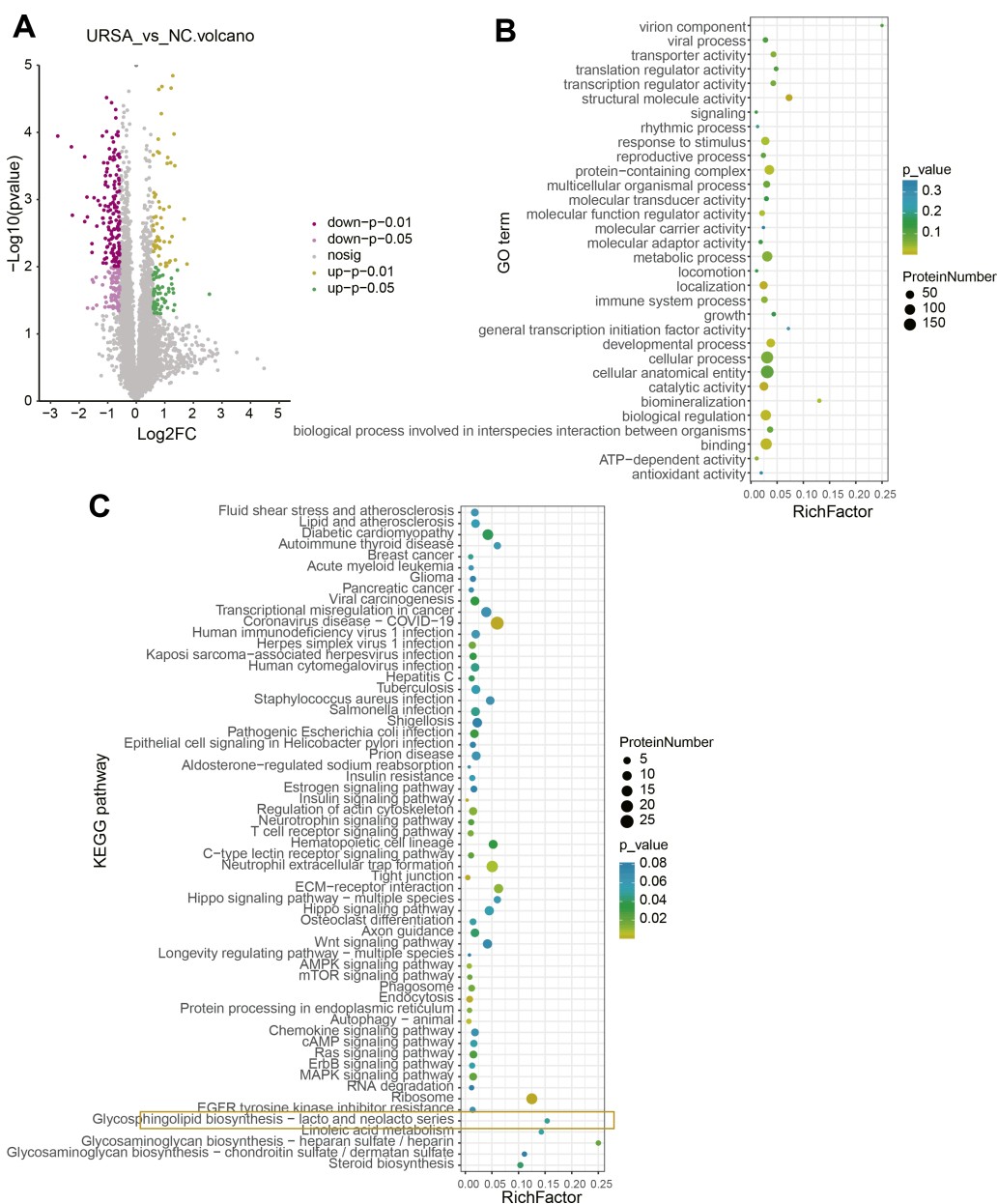

**Figure 3  Differential protein analysis between URSA and NC groups.** (A) Volcano plot of differential proteins. (B) GO enrichment analysis of differential proteins. (C) KEGG enrichment analysis of differential proteins.

choline phosphotransferase and phospholipase A (2), respectively. These genes together with other metabolites formed a complex regulation network in glycerophospholipid metabolism (Fig. 5B). Finally, the metabolite differences of the glycerophospholipid metabolism pathway between the two groups were compared by ssGSEA, the results of which demonstrated that the metabolites in the NC group were significantly higher

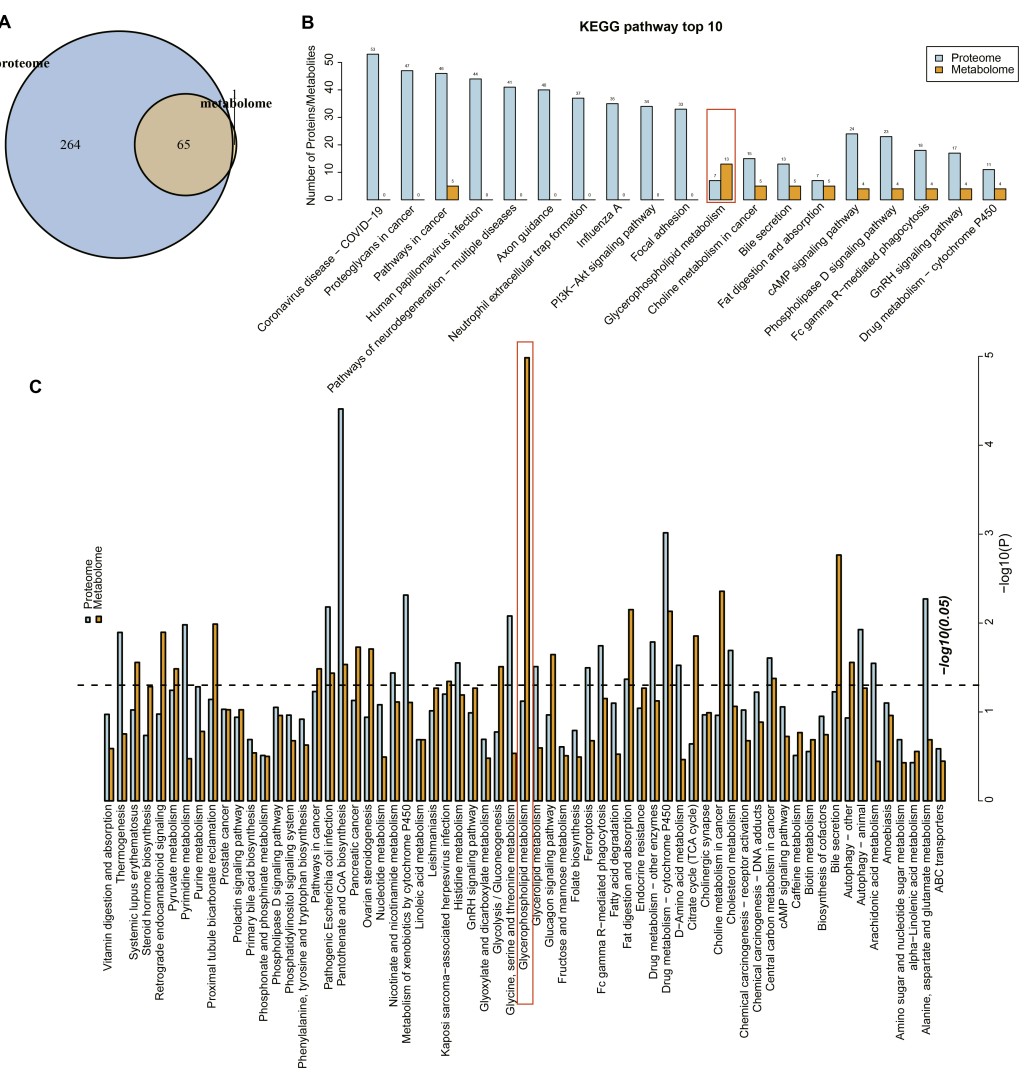

**Figure 4** **Conjoint analysis of differential metabolites and proteins.** (A) Venn plot of differential metabolites and proteins. (B) The top10 pathways of differential metabolites and proteins enriched. (C) The KEGG enrichment analysis of different proteins/metabolites in each group.

than in the URSA group (Fig. 5C). These results supported that the dysregulation of glycerophospholipid metabolism affected URSA progression.

## Expression of the key genes and experimental validation

T-hESCs with decidual induction could simulate the functional state of decidual tissue in early pregnancy stage, therefore the cells were used to further investigate the biological functions of decidual cell regulated by *PLD1*, *CHPT1*, and *PLA2G2A* in URSA. First, it was observed that compared to T-hESCs without decidualization induction, the mRNA expression levels of decidual markers *PRL* and *IGFBP1* were significantly upregulated in the decidualization group (Fig. 6A). Subsequently, qRT-PCR results showed that *PLD1*, *CHPT1* and *PLA2G2A* genes encoding key regulatory proteins were significantly upregulated in the

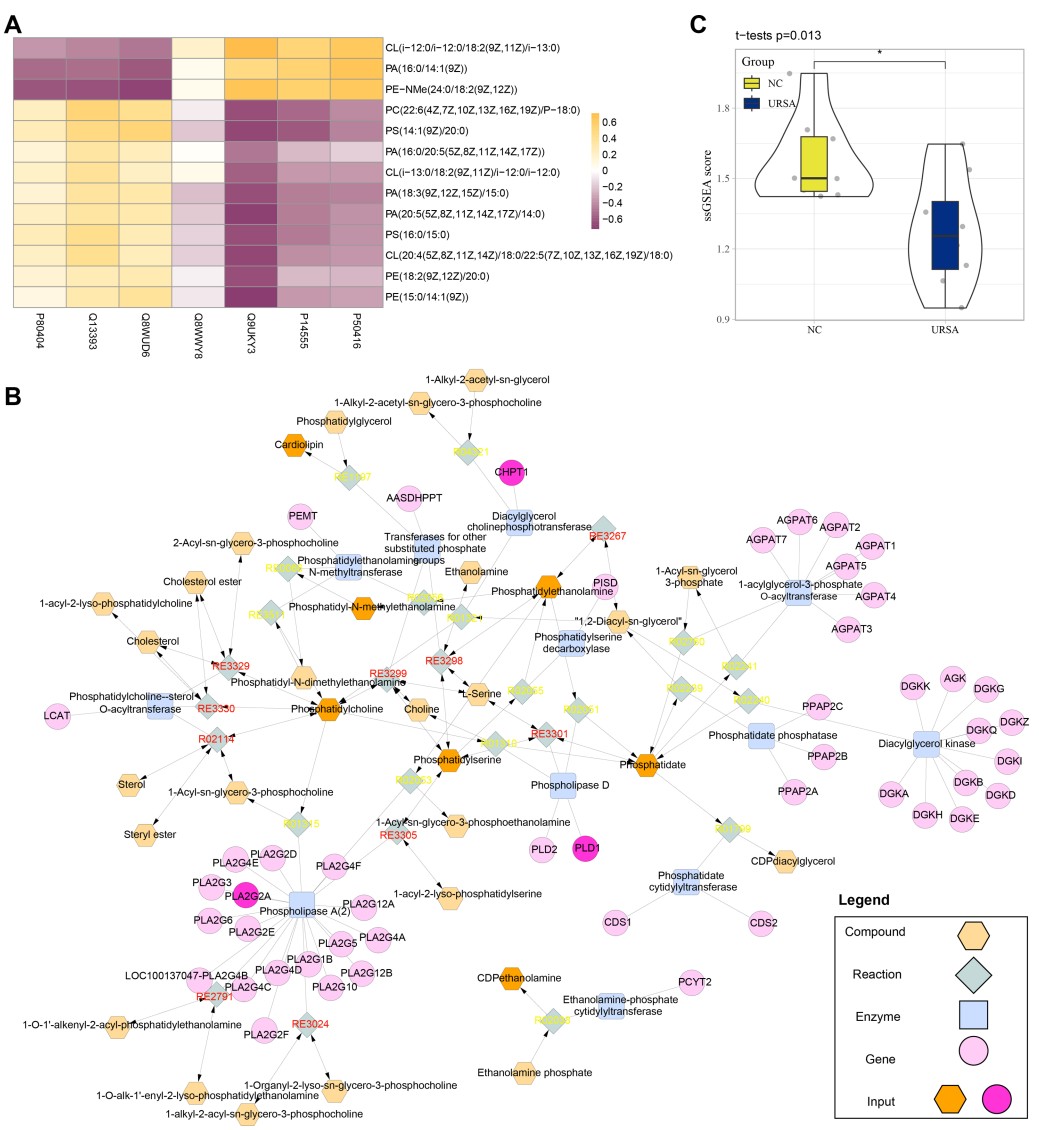

**Figure 5  Metabolic network analysis.** (A) Correlation heatmap of differential metabolites and proteins. (B) The metabolic network plot. (C) ssGSEA for glycerophospholipid metabolism pathway difference.

decidualization group (Fig. 6B). Notably, cell function assays revealed a relatively significant level of *CHPT1*, however, silencing *CHPT1* remarkably inhibited the cell proliferation rate in the si-*CHPT1* group than in the si-NC group (Fig. 6C). Furthermore, in the si-*CHPT1* group, a significant increase in early and late apoptotic cells was detected (Fig. 6D). These results indicated that these key genes functioned crucially in the occurrence and development of URSA.

## DISCUSSION

The occurrence of URSA was affected by multiple factors with unclear etiology. Comparison on the differential metabolites between the decidua of URSA and NC patients revealed

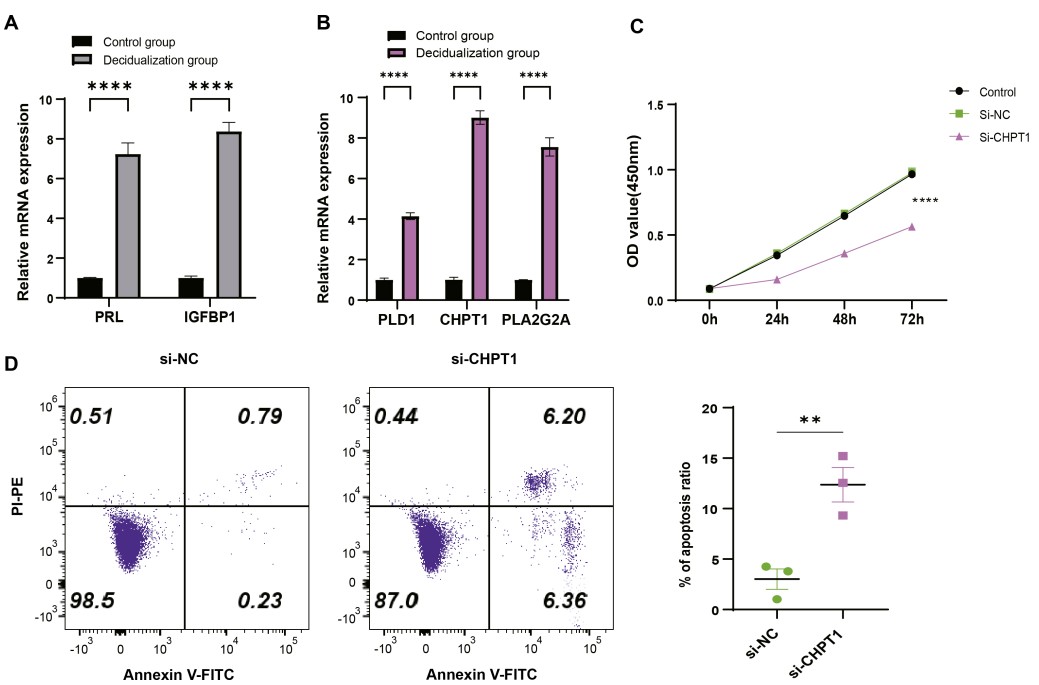

**Figure 6  Expression validation and functional assessment of key genes.** (A) Based on qRT-PCR to verify the mRNA expression levels of decidua markers *PRL* and *IGFBP1*. (B) Based on qRT-PCR to verify differential expression of key genes (*PLD1*, *CHPT1*, and *PLA2G2A*) in control and decidualization groups. (C) CCK-8-based assay to validate the effect on proliferation of decidualization induced T-hESCs after silencing *CHPT1*. (D) Flow cytometry to assess the effect on the apoptotic capacity of decidualization induced T-hESCs after silencing *CHPT1*. Control group refers to T-hESCs that have not undergone decidualization induction; decidualization group refers to the induction of decidualization of T-hESCs *in vitro*. All experiments were three independent replications. ** indicates $p < 0.01$ and **** indicates $p < 0.0001$.

that these differential metabolites were enriched in fat metabolism, in particular, glycerophospholipid metabolism and bile secretion pathways were significantly different between the two groups. This indicated a distinct alteration in metabolism in the decidua of NC and URSA patients. KEGG pathway topology also showed that abnormal fat metabolism pathways, such as steroid degradation, glycerophospholipid metabolism and steroid hormone biosynthesis, played an important role in URSA. In addition, the differential proteins shared the same pathways with differential metabolites, including glycerophospholipid metabolism, bile secretion and fat digestion and absorption. The differential proteins of P80404, Q13393 and Q8WUD6 were positively correlated with most lipid metabolites, and *PLD1*, *CHPT1* and *PLA2G2A* genes regulating the enzyme activity played a crucial role in glycerophospholipid metabolism pathway. Finally, the results of ssGSEA demonstrated that the URSA patients had a significant lower metabolism pathway score. These results strongly supported the role of abnormal glycerophospholipid metabolism in URSA progression.

During the course of pregnancy, energy metabolism consists of an anabolic phrase (at first two trimesters of gestation), which increases lipid reserves to meet the energy

consumption of fetus at third trimester, and a catabolic phase (at third trimester). This latter phrase is characterized by reduced insulin sensitivity and enhanced lipolytic activity, leading to higher concentrations of free fatty acid and circulating glucose in mother (*Zeng, Liu & Li, 2017*). Mounting evidence demonstrated that the deviation of this metabolic status might cause adverse effects on normal pregnancy. For instance, maternal pregnancy dyslipidemia is associated with preeclampsia and gestational diabetes, further increasing the risk of RSA (*Wild & Feingold, 2000*). Previous studies reported that RSA is linked to immune-related diseases, including antiphospholipid syndrome (APS) (*Kutteh & Hinote, 2014*), which produces an array of antiphospholipid antibodies, such as lupus anticoagulant (LA), anticardiolipin antibodies (aCLs) and anti-β2-glycoprotein I antibodies (aβ2GPI) to promote thrombosis and infarction and morbidity of RSA in pregnancy (*Miyakis et al., 2006*). A meta-analysis demonstrated that APS is positively correlated with RSA (*Santos et al., 2017*), suggesting that maintaining a normal phospholipid metabolism is decisive for a successful pregnancy. Glycerylphosphatide and sphingolipid are reported as the major components affecting RSA in the animal model (*Wang et al., 2021*). This conclusion was consistent with our findings of the crucial involvement of glycerophospholipid metabolism in URSA progression.

In addition, P80404, Q13393, Q8WUD6 and PE, CL, PS, PA and PC were the main differential metabolites and proteins in glycerophospholipid metabolism pathway. Another study also showed that the levels of glycerophospholipids, including the PC, PE and PS, were significantly decreased in the RSA groups in comparison to that in the normal pregnancy (NP) groups (*Wang et al., 2021*), and that these metabolites also had distinct differences between NC and URSA groups. Metabolic network map identified that the genes *PLD1*, *CHPT1* and *PLA2G2A* encoded Q13393 (Phospholipase D1), Q8WUD6 (Cholinephosphotransferase 1) and P14555 (Recombinant Phospholipase A2), respectively, together forming a complex regulating network. Phospholipase D1 (PLD1) participates in the regulation of cell growth, proliferation and protein transport (*Jenkins & Frohman, 2005*), and the expression of *PLD1* in the trophoblast cells treated by the carcinogen of benzo(a)pyrene [B(a)P] is associated with URSA (*Dai et al., 2023*). Downregulating *CHPT1* could greatly affect the synthesis of glycerophospholipids, affecting the neurobehavior of mice (*Wang et al., 2023*). Abnormal expression of *PLA2G2A* is associated with diabetes and carcinogenesis (*Zhang et al., 2022*). Overall, this study analyzed the metabolic differences between the NC and URSA, highlighting the importance of glycerophospholipid metabolism pathway in promoting URSA progression. However, there were some limitations in this study. Firstly, the clinical samples showed significant individual differences that may influence the abundance of metabolites and proteins, which increased the complexity of data interpretation. Therefore, future studies should rigorously match sample characteristics (*e.g.*, age, gender, health status, *etc.*) and validate the current findings with larger multicenter clinical samples to reduce the impact of individual differences on the findings. Secondly, the sample size of this study was limited and this may affect the statistical significance of certain results. Thus, we plan to further validate the statistical robustness of the present results by expanding the sample size and including multicenter studies. In addition, the use of data extension methods (*e.g.*, data

integration and machine learning algorithms) may further improve the predictive power of the results. Finally, current metabolomics and proteomics analyses relied on public databases and statistical significance rather than targeted assays, therefore some of the key metabolites and proteins may not be accurately identified. Hence, targeted mass spectrometry will be combined in the future to improve the accuracy of the key metabolites and proteins identified. Meanwhile, data interpretation can be further improved by establishing high-quality spectral libraries and optimizing the experimental conditions.

## CONCLUSION

In conclusion, this study applied metabolomics and proteomics analysis to systematically investigate the metabolite and protein changes in the target biological systems. Metabolomics revealed significant changes in the key metabolites (*e.g.*, phosphatidyl ethanolamine, phosphatidylcholine) involved in glycerophospholipid metabolic pathway, and proteomics further validated the regulatory proteins associated with these pathways. Importantly, the *PLD1, CHPT1*, and *PLA2G2A* genes encoding key regulatory proteins not only played important regulatory roles in glycerophospholipid metabolism-related enzymes, but also had potential effects on the proliferative and apoptotic capacities of the cells in URSA. Our findings provided important insights for elucidating the molecular mechanisms of URSA, laying a solid foundation for future study to discover potential therapeutic targets for the disease.

### Abbreviations

| | |
|---|---|
| **URSA** | Unexplained recurrent spontaneous abortion |
| **RSA** | recurrent spontaneous abortion |
| **QC** | quality control |
| **NC** | normal control |
| **LC-MS** | liquid chromatography-mass spectrum |
| **OPLS-DA** | Orthogonal Partial Least Squares Discrimination Analysis |
| **HMDB** | Human Metabolome Database |
| **TNF-α** | tumor necrosis factor-α |
| **PCA** | principal component analysis |
| **VIP** | variable importance in projection |
| **KEGG** | Kyoto Encyclopedia of Genes and Genomes |
| **EHMN** | Edinburgh Human Metabolic Network |
| **FC** | Fold change |
| **PE** | phosphatidyl ethanolamine |
| **PA** | phosphatidic acid |
| **PS** | phosphatidylserine |
| **ssGSEA** | single sample gene set enrichment analysis |

### Funding

This study was supported by Natural Science Foundation of Fujian Province (No. 2020J01986 and 2021J01236) and Fujian Provincial Finance Project (No. BPB-2023WJB1243). The funders had no role in study design, data collection and analysis, decision to publish, or preparation of the manuscript.

### Grant Disclosures

The following grant information was disclosed by the authors:
Natural Science Foundation of Fujian Province: 2020J01986, 2021J01236.
Fujian Provincial Finance Project: BPB-2023WJB1243.

### Competing Interests

The authors declare there are no competing interests.

### Author Contributions

- Yihong Chen conceived and designed the experiments, performed the experiments, analyzed the data, prepared figures and/or tables, authored or reviewed drafts of the article, and approved the final draft.
- Xiumei Zhao conceived and designed the experiments, analyzed the data, prepared figures and/or tables, and approved the final draft.
- Bei Gan performed the experiments, authored or reviewed drafts of the article, and approved the final draft.
- Leiyi Jin performed the experiments, prepared figures and/or tables, and approved the final draft.
- Juanbing Wei conceived and designed the experiments, analyzed the data, authored or reviewed drafts of the article, and approved the final draft.
- Jianying Yan conceived and designed the experiments, performed the experiments, analyzed the data, prepared figures and/or tables, authored or reviewed drafts of the article, and approved the final draft.

### Human Ethics

The following information was supplied relating to ethical approvals (i.e., approving body and any reference numbers):

Ethics Committee of First Affiliated Hospital of Fujian Medical University (Approval No. MTCA, ECFAH of FMU [2015] 084-2)

### Data Availability

The data is available in the Supplemental File and at Figshare: Chen, Yihong; Zhao, Xiumei; Gan, Bei; Jin, Leiyi; Wei, Juanbing; Yan, Jianying (2024). 16 cases of LCMSMS organization raw data and added experimental raw data.zip. figshare. Dataset. https://doi.org/10.6084/m9.figshare.27677550.v2.

## Supplemental Information

Supplemental information for this article can be found online at http://dx.doi.org/10.7717/peerj.19317#supplemental-information.

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
