# Peer review of "Metabolomics and proteomics analyses reveal the role of the glycerophospholipid metabolism pathway in unexplained recurrent spontaneous abortion"

_PeerJ, doi:10.7717/peerj.19317_

## Round 0.1 · original submission · Major Revisions

While one reviewer suggests minor revisions, the other reviewer has raised several substantial concerns that need to be thoroughly addressed. These include fundamental aspects of your methodology and data interpretation that require significant improvements. After careful consideration of your manuscript and the comments from two reviewers, I have decided that your manuscript requires major revisions before it can be considered for publication. Please provide a point-by-point response to all comments from both reviewers, with particular attention to the major concerns raised by Reviewer 1. Your revised manuscript should demonstrate substantial improvements in response to these comments.

Reviewer 1 ·

Basic reporting

1. Language and Grammar
o Grammatical errors and typos are present throughout the manuscript. I recommend that the authors seek assistance from a colleague proficient in English and familiar with the subject matter. Alternatively, AI tools can be helpful for improving grammar and clarity. For example,
o L. 129: There are two occurrences of "the" in the sentence.
o L. 134-137: The sentence begins with "And" and includes excessive use of "and." Rewrite for clarity and conciseness.
o L. 211: Replace “are showed” with "are shown."
o L. 267: The reference should end with a period, and the next sentence should start with "This conclusion ..."
o L. 279: Replace "downregulated" with "downregulation."

2. Title and Keywords
o The title should reflect both metabolomics and proteomics analyses, as these were critical to identifying the glycerophospholipid pathway. It is misleading to mention only metabolomics in the title.
o Keywords (L. 50-52): Replace "liquid chromatography-mass spectrum" with "liquid chromatography-mass spectrometry." Avoid using VIP, KEGG, and Human Metabolome Database as keywords, as they are too general.

3. Clarity in Terminology and Description
o L. 30: Replace "liquid chromatography-mass spectrometry testing" with "liquid chromatography-mass spectrometry analysis" for formality and precision.
o L. 116: Clarify the term "quality control (QC) samples." Define what QC samples are and their purpose. Also, when the authors mentioned “the quality control (QC) samples composing of each sample supernatant (20 uL)”, do they mean “samples”, and not just QC samples? Please clarify this point.
o L. 225: What are CL (i-12) and PE-NMe? Also, the text “Figure 5A” is not bolded.

Experimental design

1. Methods and Missing Details
o The manuscript lacks details regarding the proteomics analysis, including sample preparation and analysis methods.
o L. 113: Specify how many milligrams of samples were homogenized.
o L. 122: A flow rate of 2 µL/min is unrealistic for the column dimensions (100 mm × 2.1 mm i.d.). Verify and correct the flow rate.

2. Metabolomics Analysis
o Clarify how metabolites were identified. Was MS/MS conducted? Were standards used for identification?
o L. 181: Provide Q² and R² values for the OPLS-DA model to support its validity.

3. Proteomics and Metabolomics Data
o Include a complete list of all changing metabolites with their detected retention times and changing proteins.
o When discussing glycerophospholipids or other metabolites, specify which ones with which sidechains were identified.
o L. 195-196: Please provide details on the metabolites leading to the identification of caffeine metabolism. It seems strange that this metabolism would be involved.

Validity of the findings

1. Data Consistency and Interpretation
o L. 214: Address the inconsistency between metabolomics and proteomics analyses in Fig. 4C, where glycerophospholipid metabolism was enriched only statistically significantly in metabolomics.

2. Conclusion
o Since the manuscript incorporates proteomics analysis and establishes a connection between metabolomics and proteomics data, both techniques should be explicitly mentioned. Additionally, the authors should consider highlighting some of the highly associated metabolites and proteins identified in the study.

Additional comments

In this study, the authors utilized metabolomics and proteomics analyses to investigate pathways associated with unexplained recurrent spontaneous abortion. The research generated a substantial amount of data, which has the potential to provide comprehensive insights into the topic, and employed numerous statistical tools. However, the manuscript’s most significant weakness is the lack of detailed information regarding the methods and the data on the changing metabolites and proteins. The comments to help improve the manuscript were outlined above.

Reviewer 2 ·

Basic reporting

Thank you for the invitation from the editor. I have carefully read the manuscript and the main comments are as follows:
1. “Current, mounting studies reported that the URSA was closely associated with the immunological imbalance4 . “ which uses “studies” to indicate more than one report, but only one reference is cited, please add more references.
2, DATA ACQUISITION: More detailed inclusion and exclusion criteria are needed.
3、Metabolomics testing: Due to significant differences between clinical samples and insufficient sample size, this is an important limitation of this study.
4. All tissues were evaluated by pathologists. How were these evaluations conducted, and what blinding methods were employed?
5. Line 251: There is no need to re-introduce the concept of pregnancy, as it is redundant with the introduction.
6. The authors identified key genes such as PLD1, CHPT1, and PLA2G2A. It is recommended to proceed with further validation, selectively choosing some of these key genes for verification based on evidence. These genes could be knocked out or overexpressed, and then their impact on decidual cell functions, such as cell proliferation, apoptosis, and inflammatory responses, could be assessed in vitro.
7. Studies involving human samples must adhere to the ethical principles of the Declaration of Helsinki and should declare this within the text.

Experimental design

no comment

Validity of the findings

no comment

---

## Round 0.2 · Minor Revisions

Based on the comments from two reviewers, I believe your paper has merit but requires minor revisions before acceptance. Please carefully address the reviewers' comments and suggestions, and submit your revised manuscript along with a point-by-point response to the reviewers' comments.

Reviewer 1 ·

Basic reporting

The authors have addressed most of my questions satisfactorily. However, the following points still require attention:
• Title: The title appears to contain a grammatical error. It might be better phrased as: "Metabolomics and proteomics analyses reveal the role of the glycerophospholipid metabolism pathway in unexplained recurrent spontaneous abortion."
• L. 160-161: The statement “the peptides were purified” is ambiguous. Since proteomics analysis typically involves a complex mixture of proteins, it is unlikely the peptides were truly purified. Please clarify what you mean by "purified."

Experimental design

• The description of the LC-MS analysis procedure requires additional detail for reproducibility. Specifically:
o L. 121-125: Please specify how much of each sample was injected into the LC-MS instrument.
o L. 161-162: Provide details about the gradient, mobile phases, and columns used in the UHPLC system coupled with the Q Exactive mass spectrometer for the proteomics analysis.
• L. 159: The centrifugation step should be clarified by indicating the speed (g-force) and duration of centrifugation.
• L. 160: For the BCA kit, please specify the manufacturer and country of origin to ensure accuracy and reproducibility.

Validity of the findings

• Since the metabolomics community allows for different levels of metabolite identification, the methods used for metabolite identification should be described in greater detail in the article itself. This is critical for enabling the audience to evaluate the reliability and validity of the authors’ data interpretation.
• Figures and Data Availability: The raw data for the proteomics analysis is still lacking. Please provide a comprehensive list of all identified proteins, including their retention times and corresponding peptide fragments, to enhance transparency, ensure reproducibility, and support the validity of the conclusions.

Additional comments

No comment

Reviewer 2 ·

Basic reporting

Thank you for the editor's invitation again. I have carefully read the revised manuscript and the author has addressed and replied to my comments.

Experimental design

no comment

Validity of the findings

no comment

---

## Round 0.3 · accepted · Accept

The reviewers have evaluated your revised submission and are satisfied with the changes made to address their previous concerns. You have successfully clarified the scientific contribution and strengthened the methodology. I am pleased to inform you that your manuscript has been accepted for publication. The Editorial Office will contact you shortly regarding the next steps in the publication process.

Reviewer 1 ·

Basic reporting

The authors have addressed my questions/ comments satisfactorily.

Experimental design

The authors have addressed my questions/ comments satisfactorily.

Validity of the findings

The authors have addressed my questions/ comments satisfactorily.